# ACTIVE YOU: Teacher Attributes and Attitudes Predicting Physical Activity Promotion

**DOI:** 10.3390/bs13030210

**Published:** 2023-03-01

**Authors:** Erin E. Centeio, Yeonhak Jung, Darla M. Castelli

**Affiliations:** 1Department of Kinesiology and Rehabilitation Science, University of Hawai‘i, Mānoa, Honolulu, HI 96822, USA; 2Department of Kinesiology, California State University, Northridge, Northridge, CA 91330, USA; 3Department of Kinesiology and Health Education, The University of Texas at Austin, Austin, TX 78712, USA

**Keywords:** preservice teachers, schools, self-efficacy, physical activity, movement integration

## Abstract

Background: Based on the Health Belief Model, this study examined preservice teacher attributes and attitudes toward providing physical activity opportunities for children in school. Methods: A quasi-experimental design was used to collect proof of concept and feasibility data for the ACTIVE YOU intervention as part of teacher education. Conclusions: Examination of a diverse sample of preservice teachers during their fieldwork revealed that those who engage in healthy behaviors and had positive attitudes toward physical activity in schools are more likely to take action and promote physical activity for their students.

## 1. Introduction

Among children, single bouts of physical activity [1] and regular participation [2] positively relate to selective academic outcomes, as healthier children are more ready to learn [3]. Although the specific physical activity characteristics, such as the type and intensity that directly influence academic achievement may be debatable, the health-protective benefits of improved mental and social-emotional health of children justify its inclusion in the school curriculum [4]. Movement integration (MI), the integration of physical activity into classroom learning activities, is generally positively perceived by teachers [5,6]. When MI or physical activity is implemented effectively, it can improve student behaviors, increase time on task during academic learning [7], and increase the likelihood of having a positive class climate [8]. Schools provide an opportunity for children to learn the importance of physical activity in a structured and safe environment where knowledge and skills can be developed.

Although schools are an ideal place to promote physical activity, few teachers are prepared by their teacher education programs to do so. Novice teachers have many responsibilities, including their role as health promoters [9,10]. Even with training, teacher rankings of the importance of health promotion are low [11,12], and the influence of personal health behaviors influence attitudes, biographical characteristics, and self-efficacy toward leading physical activities in school has only been cross-sectionally examined among a few samples of preservice teachers [13,14] or as cohort research [15,16]. As the baseline assessment in a children’s movement course, we administered a standardized health-related fitness assessment to 133 individuals in their third year who were majoring in education. The assessment was valid for ninth-grade students [17]. The mean score for the preservice K-6 teachers was 54% correct responses, thus demonstrating the lack of health-related fitness knowledge among future teachers.

In response to such concerns, we developed an intervention, ACTIVE YOU, to address the gaps in teacher education. Specifically, we sought to inspire cues to action and build self-efficacy to provide physical activity opportunities for children in schools. This study seeks to confirm the feasibility and proof of concept of the teacher education intervention, ACTIVE YOU, which focuses on developing health promotion skills among preservice teachers during their fieldwork.

### 1.1. Theoretical Framework: Health Belief Model

The Health Belief Model (HBM; Figure 1) links multiple constructs to explain why humans embrace some healthy behaviors while avoiding others [18]. In this study, we have applied the constructs of HBM to the lives of teachers, who must simultaneously consider their health and wellness and that of their students. The HBM is grounded in the constructs of perceived susceptibility, perceived severity, perceived benefits/barriers, cues to action, and self-efficacy. A teacher’s personal beliefs surrounding their health risk (e.g., a sedentary lifestyle, obesity, or poor eating habits) typically intensify with age [19]. Teachers must be aware of their own habits to conscientiously reduce personal health risks, considering that health is a continual work in progress affected by daily behaviors. Therefore, the perceived severity or the seriousness of a potential health condition would also play a role in a teacher’s attitudes and actions.

### 1.2. Teacher Education: Beliefs and Self-Efficacy

According to Bandura’s Social Cognitive Theory (1997), self-efficacy is the “belief in one’s capacity to organize and execute the course of action required to produce given attainments” [20] (p. 3). Teacher education encourages preservice teachers to be efficacious in their ability to enhance student learning because we know that past experiences in physical education and physical activity shape preservice teachers’ beliefs and influence what and how they teach [21]. The professional socialization process is designed to build self-efficacy through mastery experiences, vicarious experiences, verbal persuasion from teacher educators, and verbal persuasion from cooperating teachers [22].

The formulations of such beliefs through this contemplation can trigger actions. As presented in Figure 1, an event or cue (e.g., vicarious experiences—watching a fourth-grade teacher model how they integrate MI into a lesson; mastery experiences—the preservice teacher integrating MI into a lesson with fourth-grade students) can often initiate a behavioral response. Specific experiences in teacher education are designed as a source of cues to action [9]. For example, the mastery experience included in ACTIVE YOU are authentic case studies. When the case studies focus on personal health-protective behaviors, such as participation in physical activity, the reflective practice can build preservice teacher efficacy toward their capacity to model health practices for their students. Ultimately, the teacher must have the confidence to assert their actions. Those who believe they can influence the events, such as improving their health or that of their students, will behave in a goal-directed manner [23].

### 1.3. Teachers as Health Promoters in Schools

Targeting multiple points of intervention, the Comprehensive School Physical Activity Programs (CSPAP) framework creates a culture of consistent messaging, promoting opportunities for all to engage in positive and healthy physical activity during physical education, before and after school, during the school day, and with family and community members on school grounds [24]. A classroom teacher plays a pivotal role in implementing the school day portion of the CSPAP, partly by modeling but also by facilitating and supervising physical activity opportunities. Therefore, teacher education learning activities that introduce classroom educators to effective ways to incorporate MI and physical activity across the school day may benefit students and teachers [24,25].

Whether formally a school champion or simply a teacher who understands that healthier children learn better, positive attitudes and self-efficacy are valuable dispositions that influence the effectiveness of such initiatives [26,27]. Accordingly, this study sought to identify the relationship between preservice teacher attitudes and self-efficacy related to promoting physical activity within the school environment from an HBM perspective.

If teacher education programs are to effectively prepare teachers as health promoters who are likely to model and offer physical activity opportunities for their students, the learning experiences should be aligned with best practices and the theoretically and empirically supported literature [28]. As such, this study examined the association between preservice teacher characteristics, such as Body Mass Index (BMI) and daily physical activity participation, and their likelihood of offering physical activity opportunities in the school setting. Uniquely positioned in a preservice teacher education program at one US university, these future teachers were enrolled in a required course focused on children’s movement content. The state where this study took place has mandated courses for preservice teachers to provide physical education subject matter content and to introduce the whole-of-school approach using models such as CSPAP. Accordingly, this study sought to determine the feasibility and proof of concept of the ACTIVE YOU intervention to meet this state mandate and increase the likelihood that future teachers would provide physical activity opportunities during the school day for their students.

## 2. Materials and Methods

Using a quasi-experimental design in a 16-week teacher education course, we examined the relationships between personal attributes of BMI and daily physical activity participation on attitudes and self-efficacy toward providing physical activity opportunities for children among preservice teachers. We also examined the predictors of physical activity promotion. Findings from this study could provide proof of concept and support for continued development and full integration of the ACTIVE YOU curriculum, which has been organized into the Template for Intervention Description and Replication (TIDieR checklist [29], see Table 1). To date, the protocol has not yet been registered as a clinical trial.

### 2.1. Participants

After an Institutional Review Board approval was secured, 265 preservice teachers enrolled in a teacher education certification program volunteered to share their assignments completed during the university course. The researchers collected the data from the courseware and wearable device software after completing the course, with grades posted and course credit awarded. One student declined to provide the requested information for research purposes.

### 2.2. Instruments and Measures

Data were collected using wearable technology, self-reported perceptions, lesson reflections, fieldwork, and coursework artifacts over 16 weeks of fieldwork by preservice teachers. Two sets of valid, reliable surveys were administered at the beginning and end of the course: (a) past perceptions and experiences and (b) perceptions about fieldwork and working with children in schools.

#### 2.2.1. Daily Physical Activity Measures of Self

The personal daily physical activity level of the preservice teachers enrolled in the course was objectively measured using accelerometers and subjectively measured through logs and reflections. Step counts were assessed using an ActiGraph GT3X (ActiGraph, Pensacola, FL, USA), which uses a solid-state triaxial accelerometer to measure motion data on multiple axes. The accelerometers were initialized to collect data using 5-s epoch lengths to capture the sporadic, intermittent nature of participant physical activity [30]. The participants were also required to keep a physical activity log that reflected their daily participation. Both measurements were integrated into the ACTIVE YOU intervention as course requirements intended to increase physical activity participation.

#### 2.2.2. Biographical Questionnaire of Physical Self and Self-Perceptions of Ability

Self-reported biographical information, including participants’ age, gender, race, height, weight, and year in the teacher education program, was attained via survey through the campus courseware. BMI was calculated by dividing one’s weight in kilograms by the square of one’s height in meters. Participant’s perceptions of physical activity competence and past physical activity experiences were assessed with five items and four items, using two valid, reliable, 4-point Likert scales (1 = strongly disagree to 4 = strongly agree) with acceptable internal consistency (>0.82 and >0.86) [31]. Example items include “I am capable of participating in multiple sport activities” and “I enjoyed my physical education experience when I was in school”.

#### 2.2.3. School Physical Activity Promotion Competence (SPAPC)

The School Physical Activity Promotion Competence (SPAPC) [9,10] is a valid, reliable instrument used to determine self-efficacy toward promoting physical activity in a school setting. The instrument contains 15 items and uses an 8-point Likert scale (0 = no skills to 7 = many skills) with acceptable internal consistency (>0.95). Example items include “I am comfortable integrating physical activity into classroom lessons”.

#### 2.2.4. School Physical Activity Promotion Attitudes (SPAPA)

The School Physical Activity Promotion Attitude (SPAPA) [13,14] was used to measure attitudes toward promoting physical activity in a school setting with nine items using a 4-point Likert scale (1 = strongly disagree to 4 = strongly agree) with acceptable internal consistency (>0.83). Example items include “Elementary classroom teachers should provide physical activity for students daily as part of the school day”. The sum of each instrument was included in the statistical analysis.

#### 2.2.5. Power Analysis

A priori power analysis was computed using G*Power to estimate the sample size needed for calculating hierarchical and logistical regression analyses based on the correlations and effect size reported in a meta-analytic review [32]. The calculation was based on a medium effect size (f = 0.25), six groups, and four covariates. Results showed that a total of 179 participants were required to achieve 80% statistical power at *α* = 0.05. However, the final decision for sample size was *n* = 200 in consideration of potential attrition.

### 2.3. Data Collection Procedures

Participants were recruited from six course sections offered by a teacher education program after permission for intervention was obtained from the IRB and course instructors. Among the sections, three were randomly assigned to treatment groups and three were assigned to serve as control groups over one academic year. As part of the planned learning experiences within the course, participants were provided with formal instruction on the benefits of physical activity participation, which included familiarization with the instruments (e.g., accelerometer). All participants received a daily physical activity log and an accelerometer, which they were asked to wear for 14 days (7 days early in the semester and 7 days later in the semester). Before delivering physical activity content, the participants were asked to complete the online surveys before the end of the first week of the course. The treatment fidelity was confirmed through weekly lesson content observations, lesson artifacts review (e.g., presentation slides and assignments on courseware), and having one teacher deliver the intervention content to all course sections. All lessons were aligned with the HBM constructs (Figure 1), with the *likelihood of action* being measured as participation in and promoter of physical activity.

### 2.4. Statistical Analysis

This study used SPSS 25.0 software to analyze and process the data. Data screening was conducted before any statistical analyses were performed. For survey instruments, skewness and kurtosis values were used to confirm the normality assumption, while Cronbach alpha coefficients were used to assess internal consistency for validity [33]. Descriptive statistics were also calculated, including means (M), standard deviations (SD), and relative frequencies for continuous and categorical data. Based on two groups, including the healthy BMI group and the unhealthy BMI group, continuous variables (i.e., age, BMI, daily physical activity level) were compared using independent *t*-tests. The relationships between the BMI classifications, daily physical activity levels, and survey scores were examined using correlations. A hierarchical regression analysis was conducted to determine the likelihood of having positive attitudes and self-efficacy toward promoting physical activity in a school setting. The hierarchical regression analysis was selected to account for modifiable (e.g., attitudes, self-efficacy) and non-modifiable (i.e., year in teacher education program and race) variables. The hierarchical regression model was selected over other regression analyses because it examines the relationships between independent variables and a dependent variable after controlling for the effects of some non-modifiable variables. Logistic regression was used to predict the likelihood of having daily physical activity levels and perceived physical activity competence between the healthy BMI group and the unhealthy BMI group. Logistic regression was selected for two reasons: (a) the dependent variable was categorical and (b) the researchers wanted to estimate an odds ratio for these variables. The statistical significance level of all indicators was set as *p* < 0.05.

## 3. Results

A total of 265 participants were enrolled in the study over the academic year; however, only 233 participants were included in the final statistical analysis because they provided 14 days of physical activity measures and had completed all survey instruments. Thirty-two participants were excluded due to the absence of physical activity measures, incomplete surveys, or excessive missing data (i.e., missing more than 50% of the total values).

Table 2 displays the participant demographic characteristics. A total of 233 preservice teachers (*n* = 211 females; *Mage* = 20.17, *SD* = 2.60) met the inclusion criteria. The participants were 55% White, 31% Hispanic, 8% Asian, 4% Black, and 2% who reported multiple races/ethnicities. The sample’s experience classifications demonstrated 17% first-year, 35% second-year, 28% third-year, 13% fourth-year, and 7% fifth-year preservice teachers in the teacher education program. For assessment of health behavior, steps per day (*M* = 7880.32, *SD* = 2861.66) fell below the daily standard of 10,000 steps [34], while the BMI (*M* = 22.38, *SD* = 3.68) fell within a healthy standard for this age group [35]. According to the BMI classification, participants were identified as underweight (*n* = 21; 9%), healthy (*n* = 168; 72%), overweight (*n* = 36; 16%), and obese (*n* = 8; 3%). In the subgroup comparison, 168 participants were classified in the healthy BMI group (*M*_BMI_ = 21.42, *SD* = 1.72) and 65 participants were classified in the unhealthy BMI group (*M*_BMI_ = 24.86, *SD* = 5.74). In addition, 188 participants were classified in the low active group (*M*_Steps_ = 6930.83, *SD* = 1794.43) and 45 participants were classified in the high active group (*M*_Steps_ = 12234.12, *SD* = 2812.02). When considering daily steps by the BMI classification, and BMI by the active classification, there was no significant difference in daily physical activity level, including steps (*p* = 0.19), between the unhealthy BMI group and healthy BMI group, and no significant difference in BMI levels (*p* = 0.52) between the low physically active group and the high physically active group.

### 3.1. Correlation Table Participant Characteristics, BMI, and Physical Activity Level

Table 3 reports a negative correlation between BMI and attitudes toward promoting physical activity in a school setting (*r* = −0.17, *p* = 0.009). At the same time, there was a significantly positive relationship between daily physical activity levels and perceived physical activity competence (*r* = 0.28, *p* < 0.001) and attitudes (*r* = 0.15, *p* = 0.04) toward physical activity. Attitudes toward physical activity were significantly correlated with past physical activity experience (*r* = 0.33, *p* < 0.001) and perceived competence (*r* = 0.29, *p* < 0.001). Self-efficacy toward physical activity was significantly related to past physical activity experiences (*r* = 0.21, *p <* 0.001) and perceived physical activity competence (*r* = 0.24, *p* = 0.003).

### 3.2. Hierarchical Regression Analysis

The hierarchical regression model examined personal characteristics on attitudes and self-efficacy toward promoting physical activity in school (Table 4). Regarding attitudes, the hierarchical regression analysis revealed that perceived previous physical activity experience and physical activity competence in the first step and predictors of daily physical activity level and BMI level in the second step accounted for 13% and 16% of the variance and were statistically significant, *F* (4,229) = 17.56, *p* < 0.001, η^2^ = 0.13 and *F* (6,227) = 8.82, *p* < 0.001, η^2^ = 0.16, respectively. Controlling for the year in the program and race, previous physical activity experiences (*β* = 0.24, *p* = 0.01), perceived physical activity competence (*β* = 0.17, *p* = 0.001), and daily physical activity level were positive predictors (*β* = 0.15, *p* = 0.042) of teachers’ attitudes toward physical activity promotion, while BMI was a negative predictor (*β* = −0.15, *p* = 0.015). 

Additionally, analysis of self-efficacy toward physical activity using hierarchical regression, controlling for the year in the teacher education program and race, revealed that perceived physical activity competence and previous physical activity experience in the first step and predictors of daily physical activity levels and BMI in the second step accounted for 7% and 8% of the variance, which were significant, *F* (4,229) = 5.21, *p* = 0.002, η^2^ = 0.07 and *F* (6,227) = 2.98, *p* = 0.008, η^2^ = 0.07, respectively. Only perceived physical activity competence in the first step (*β* = 0.18, *p* = 0.012) and the second step (*β* = 0.19, *p* = 0.014) was a predictor of self-efficacy toward physical activity. No other significant predictors of self-efficacy toward promoting physical activity existed. Participants with higher perceived physical activity competencies and positive previous physical activity experiences were more likely to have positive attitudes and higher self-efficacy toward promoting physical activity in the school setting. Further, those with higher BMI or lower physical activity levels were less likely to have positive attitudes.

### 3.3. Logistical Regression Analysis

The logistic regression model compared teachers’ characteristics according to the BMI classification (i.e., healthy BMI group versus unhealthy BMI group) and active classification (i.e., high active or low active group) while controlling for covariates, including race and year in the program (Table 5). Significant predictors of BMI classification in the model were daily step counts (*β* = −0.01, *p <* 0.001) and perceived physical activity competence (*β* = −0.18, *p =* 0.017). Although participants in the unhealthy BMI group were only 1% less likely to have daily step counts (OR = 0.99; 95% CI = 0.99–0.99), they had 17% less likely to have perceived physical activity competence (OR = 0.83: 95% CI = 0.71–0.96) compared to the participants who were in the healthy BMI group. In addition, only perceived physical activity competence was a significant predictor for active classification (*β* = −0.37, *p <* 0.001). Participants in the low active group were 31% less likely to have perceived physical activity competence (OR = 0.69: 95% CI = 0.58–0.82) than those in the high active group. The results indicated that the participants in the unhealthy BMI group or the low active group were less likely to have their physical activity competencies related to teachers’ attitudes and self-efficacy toward promoting physical activity in school settings.

## 4. Discussion

We hypothesized that several well-established psychosocial determinants, such as attitudes and self-efficacy toward promoting physical activity, would be related to teachers’ health behaviors, such as personal BMI and daily physical activity levels. This study identified the personal health behaviors influencing teachers’ attributes to promote physical activity engagement for their future students in the school environment. In the present study, a higher BMI level was negatively related to teachers’ attitudes toward promoting physical activity, while a higher daily physical activity level was positively related to teachers’ attitudes. These findings indicated that preservice teachers who had a high prevalence of healthy behaviors would likely positively influence their physical activity promotion in schools. This study found that just under one-third of the individuals entering into the field of teaching were at risk for health issues according to the classification of their BMI [34] and 80% of them were at risk of health issues because of the number of days not meeting the recommended daily step count [35]. Such unhealthy behaviors could be exacerbated over time, possibly due to reduced physical activity and increased sedentary activity levels related to employment as a full-time teacher [36]. Given the decline in physical activity participation in later life, other health risks, and the risk of developing as overweight/obese, the findings raise concerns regarding the potential influences of teachers’ health behavior and their perception of their future students.

These are valuable findings for two reasons. First, teachers’ health and well-being are essential, as healthier teachers are more likely to provide ample physical activity to produce academically successful students [37,38]. Additionally, there is a direct relationship between the amount of physical activity educators participate in and their willingness to promote physical activity for their students [39]. Second, the findings confirmed a bi-directional relationship between health; healthier teachers teach better and healthier students to learn better [40] because unhealthy behavior increases health problems (i.e., obesity and diabetes), which might play a significant role in interfering with children’s ability to learn in a school setting [3]. Furthermore, teachers, staff, and community members should work collaboratively to create a school setting to ensure that each student is emotionally and physically healthy.

### 4.1. Teacher Education and Preservice Teachers’ Health

Since unhealthy behavior (i.e., higher BMI level and lower physical activity participation) was negatively related to a teacher’s attitudes toward providing physical activity opportunities, teacher education needs to include coursework addressing health risks and strategies to promote physical activity in schools. Although preservice teachers with better health behaviors were more likely to have positive attitudes toward promoting physical activity in school, they lacked self-efficacy as a sense of confidence in personal physical activity skills and knowledge [41]. Low self-efficacy can decrease physical activity participation and reduce the teachers’ likelihood to promote physical activity with the students [42]. Our findings also confirmed that self-efficacy was significantly correlated with perceived physical activity competence and previous physical activity experience. The logistic regression model showed that preservice teachers in the unhealthy behavior group (i.e., BMI and active classification) had lower perceived physical activity competence than those in the healthy group, suggesting that those preservice teachers might have less confidence in promoting physical activity in their future students than those who are more active.

Findings from this study suggest that teacher education coursework in the United States should holistically integrate lessons that consider student and teacher health and wellness, similar to current global efforts [40]. Without an awareness of the potential impacts, including personal health behavior (i.e., physical activity and BMI), teachers’ attitudes, and self-efficacy, educators cannot create a healthy learning environment for students. As the present study indicated, preservice teachers who regularly engaged in physical activity had positive attitudes toward offering similar experiences to children in school. Therefore, ensuring that future educators are aware of how personal habits and competency may later influence student habits is essential.

### 4.2. Teacher Education and Preservice Teachers’ Physical Activity Promotion

Teachers are ideal health promoters, as healthier students are more ready to learn. However, few teachers are directly prepared with strategies first to develop a life-work balance and routines for their own well-being but second to also apply specific strategies for the physical activity promotion of their students. Accordingly, the field experience during the course where these data were collected was predicated on the notion that schools should provide opportunities to personally engage in physical activity, as well as to acquire content knowledge about movement concepts and motor skills, and to have exposure to instructional methods to deliver safe physical activity opportunities for children. The course content did not exclusively promote physical activity over motor skill development, both were equally endorsed. Further, the preservice teachers in this study were required to participate in a weekly field experience in an elementary school setting for approximately ten hours while assisting physical education and classroom teachers. It appeared that such experiences contributed to helping preservice teachers identify their own level of competence, efficacy, and attitudes toward facilitating student physical activity.

This study found that participation in physical activity related to preservice teacher attitudes and there was carry-over into the field experiences; therefore, this study suggests that individuals should be afforded opportunities to increase content knowledge within the classroom during teacher education. Moreover, applying this learning in an authentic context is essential to increase their perceived competence and engagement in physical activity. Since most preservice teachers in this sample exhibited positive attitudes and higher self-efficacy toward physical activities, it is probable that they will increase the likelihood of participating in and promoting health-enhancing behaviors, such as physical activity, when presented with authentic, in-school experiences.

## 5. Conclusions

Although the next step is an efficacy trial, sufficient evidence exists to include wearable devices, acknowledgment of previous physical activity experiences, teacher case studies, and specific strategies for physical activity promotion in schools (e.g., the CSPAP model). This study discovered that some individuals entering the teaching field have health risks, given the classification of their BMI and physical activity level. Preservice teachers with health risks may assume that health is an unimportant or devalued topic in schools. Given their sedentary, inactive lifestyles, today’s children are also at risk for health issues that can influence their health in relation to their academic achievement [43]. Since schools are an ideal place to offer physical activity opportunities, teachers need to be prepared and have positive attitudes and self-efficacy toward promoting physical activity. Further investigation is needed to evaluate the sustainability potential of this intervention, ACTIVE YOU, by following the preservice teacher to their full-time employment. Only then will we understand the effects of the teacher education coursework.

## Figures and Tables

**Figure 1 behavsci-13-00210-f001:**
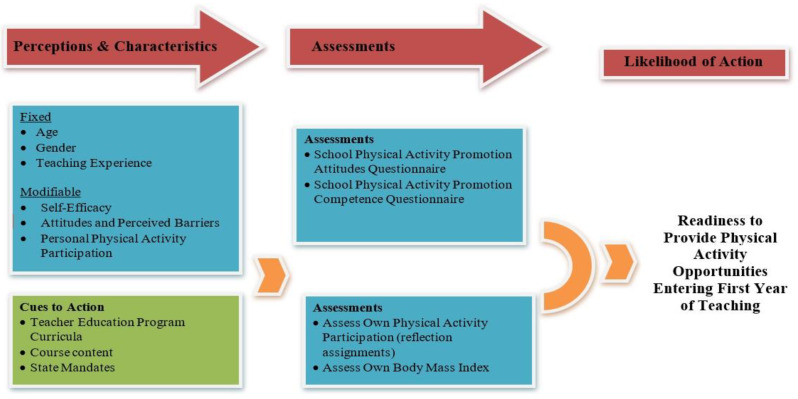
Integrating the Health Belief Model framework into the ACTIVE YOU preservice teacher intervention.

**Table 1 behavsci-13-00210-t001:** TIDieR Intervention Checklist.

Item No	Item
Brief Name	ACTIVE YOU—Preparing teachers to be health promoters in schools by building self-efficacy and reflecting on their participation in physical activity as a health-protective factor.
Why?	Reasons why this intervention matters at this time: Self-efficacy is specific to the task—preservice teachers need positive physical activity experiences to be models for students. Healthier teachers have better attendance, which increases the likelihood of student success. Preservice teachers lack health-related content knowledge. State mandates require teachers to facilitate student participation in physical activity during the school day.
What?	The intervention group participated in learning activities designed to increase personal and student participation in physical activity: (a) measurement, evaluation, and reflection on their physical fitness and physical activity participation; (b) planning and teaching a movement integration (MI) activity in the classroom using a task card from Promoting Physical Activity and Health in the Classroom (Pangrazi, Beighle, and Pangrazi, 2021); (c) vicarious experience a whole-of-school-approach through fieldwork; and (d) conducting a model teach with elementary school children and engaging in cognitive coaching debriefing with the cooperating teacher and teacher educator.
Who provided?	A single faculty member from one large university in the southern portion of the United States delivered the content across the six sections for the feasibility and proof of concept study. In the fieldwork, there were 15 co-teachers trained in mentoring and cognitive coaching.
How?	The students met on campus for 180 min per week in a classroom with an attached open space where they could participate in MI and were in a public school where they co-taught in schools for 10 h per week.
Where?	The primary delivery of the content occurred on the university campus, with the preservice teachers designing and delivering their lessons in public elementary schools in a predominantly metropolitan area.
Tailoring?	The only individualization or tailoring that occurred was accommodation by the preservice teachers to allow ALL K-6 students in their fieldwork to participate in the physical activity lesson they created.
Modifications?	In parallel with the 2018 Physical Activity Guidelines for Americans, each study participant was expected to engage in 150 min of moderate-to-vigorous physical activity each week.
How well?	Fidelity to the treatment was confirmed in multiple ways: (a) the same instructor delivered the content in all sections; (b) the physical activity logs and reflections were submitted electronically through the courseware website and were time-stamped; and (c) the same prompts were released at the same time for the fall and spring sections. There were no disruptions in the schedule (e.g., school or class cancellations).

**Table 2 behavsci-13-00210-t002:** Participant Characteristics.

Variables	(*n* = 233) (M [SD])
Age (years; M [SD])	20.17 [2.60]
Gender (female, *n* [%])	211 [90.55]
Race (*n* [%])	
White	129 [55.36]
Hispanic	73 [31.33]
Asian	19 [8.15]
Black	8 [3.43]
More than 1 race	4 [1.71]
Year in Teacher Education (*n* [%])	39 [16.73]
1st year	84 [36.05]
2nd year	65 [27.89]
3rd year	30 [12.87]
4th year	15 [6.43]
5th year	22.38 [3.68]
BMI ^1^ (kg/m^2^; M [SD])	
Physical Activity Level (steps; M [SD])	7880.32 [2861.66]

^1^ BMI = Body Mass Index; physical activity level = average number of steps per day.

**Table 3 behavsci-13-00210-t003:** Correlation Table.

Variable	1	2	3	4	5	6
BMI (kg/m^2^)	-					
Physical Activity Level	0.01	-				
Past PA Experience	−0.08	0.06	-			
Perceived PA Competence	−0.11	0.28 ^b^	0.44 ^b^	-		
Attitudes toward PA Promotion	−0.17 ^b^	0.15 ^a^	0.33 ^b^	0.29 ^b^	-	
Self-Efficacy toward PA Promotion	−0.07	0.06	0.21 ^b^	0.24 ^b^	0.27	-

Abbreviations: BMI = Body Mass Index; PA = physical activity; PA Level = daily physical activity; ^a^ *p* < 0.05, ^b^ *p* < 0.01.

**Table 4 behavsci-13-00210-t004:** Hierarchical Regression Results.

	Predictor	R^2^	β	t	*p* Value
First step	**Attitudes toward PA**				
	Previous PA Experience	0.13	0.24	3.62	0.010
	Perceived PA Competence		0.17	2.60	0.001
Second step	Previous PA Experience	0.16	0.25	3.27	0.001
	Perceived PA Competence		0.14	1.97	0.049
	Daily PA Level (steps)		0.15	2.04	0.042
	BMI (Kg/m^2^)		−0.15	−2.45	0.015
First Step	**Self-Efficacy toward PA**	0.07			
	Previous PA Experience		0.10	1.40	0.16
	Perceived PA Competence		0.18	2.54	0.01
Second step	Previous PA Experience	0.08	0.09	1.32	0.19
	Perceived PA Competence		0.19	2.47	0.01
	Daily PA Level (steps)		0.08	1.23	0.22
	BMI (Kg/m^2^)		0.00	0.01	0.99

Abbreviations: BMI = Body Mass Index; PA = physical activity. Controls are school year and race.

**Table 5 behavsci-13-00210-t005:** Logistic Regression Results.

Predictor	Β	SE	Odds Ratio	95% CI	*p* Value
**BMI Classification**					
PA level (steps)	−0.01	0.01	0.99	(0.99–0.99)	0.001
Perceived PA Competence	−0.18	0.08	0.83	(0.71–0.96)	0.017
Previous PA Experience	0.11	0.08	1.11	(0.95–1.29)	0.17
Attitudes toward PA	−0.04	0.06	0.99	(0.85–1.07)	0.54
Self-Efficacy toward PA	−0.01	0.01	0.96	(0.97–1.01)	0.44
**Daily PA Level**					
BMI (Kg/m^2^)	0.01	0.05	1.01	(0.91–1.12)	0.80
Perceived PA Competence	−0.37	0.08	0.69	(0.58–0.82)	0.001
Previous PA Experience	0.14	0.08	1.14	(0.97–1.34)	0.09
Attitudes toward PA	0.01	0.01	1.00	(0.98–1.01)	0.90
Self-Efficacy toward PA	−0.2	0.07	0.98	(0.85–1.12)	0.88

Abbreviations: BMI = Body Mass Index; PA = physical activity; CI = confidence interval. Controlled for the year in the teacher education program and race.

## Data Availability

The data presented in this study are available on request from the corresponding author. The data are not publicly available to protect the identity of study participants.

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
