# Peer review of "ACTIVE YOU: Teacher Attributes and Attitudes Predicting Physical Activity Promotion"

_behavsci, 2023, doi:10.3390/bs13030210_

Round 1
Reviewer 1 Report
This an excellent research work that deserves to be published
I want to follow up on a few points:
1- Adding keywords: physical activity; preservice teacher.
2- Regarding Table 1. TIDieR Intervention Checklist is somewhat ambiguous. It is better to re-simplify it.
Author Response
We would like to thank the review for their supportive and constructive comments. We appreciate you dedicating valuable time to as a service to our profession. Here are our responses to your suggestions:
1- Adding keywords: physical activity; preservice teacher. - Important, thank you. We have added the keywords, as suggested.
2. 2- Regarding Table 1. TIDieR Intervention Checklist is somewhat ambiguous. It is better to re-simplify it. - We have revised the TIDieR intervention checklist to provide more intervention specific detail in some sections, while exclusing extraneous text about procedures. Reviewer #2 requested more detail, so we attempted to provide a consistent amount of valuable information in each section of the checklist.
Reviewer 2 Report
This study was conducted under the theme of "Teacher Attributes and Attitudes Predicting Physical Activity Promotion," which affirms the research importance of this topic. The following are some reminders that the authors would like to add to the value and contribution of this study:
(L36): "ACTIVE YOU" is the main framework for the implementation of this study, and the authors have already presented a framework for "ACTIVE YOU" in this paper, but it would be helpful to add relevant theories to the planning and implementation of "ACTIVE YOU." (L57): The authors have already presented the framework of "ACTIVE YOU" in this paper, but if we can add the relevant theories and descriptions of the planning and implementation of "ACTIVE YOU," it will be more practical for this study to provide a reference for related researchers.
(L57): The authors hope to strengthen the basis of this paper by giving examples, but the existing questions alone cannot support the importance of this paper, so it is suggested that the authors can cite more conclusions from existing related studies to support the importance of this study, which can enhance the value of this study more obviously.
(L114): The authors present the TIDieR Intervention Checklist in a tabular format, but
However, the authors could further explain the correspondence between "HOW" and "What," and the materials and procedures used in this study are the focus of the study implementation. For example, the authors used "wearables devices (accelerometers)" as the source of data collection. What is the specific information that can be collected, and what is the baseline of this study? These are the primary basis for presenting the effectiveness of this study.
(L168): The Data Collection Procedures section is an essential part of this study. According to the information provided by the authors, the participants were asked to wear for 14 days (7 days early in the semester and 7 days later in the semester).
The authors provide information that the participants were asked to wear for 14 days (7 days early in the semester and 7 days later in the semester). Can the authors further explain why this arrangement and the detailed implementation procedures of the study, and at the same time, the linkage between this study and the course content can also be explained through understanding the intervention of this study will help the reader to have a more detailed understanding of this study, we look forward to seeing complete content from the authors.
Author Response
We thank you for your careful and thoughtful review of our paper. We believe your recommendations will strengthen the manuscript and provide enhanced clarity and detail about the intervention development and design. Here are our responses to recommendations:
- (L36): "ACTIVE YOU" is the main framework for the implementation of this study, and the authors have already presented a framework for "ACTIVE YOU" in this paper, but it would be helpful to add relevant theories to the planning and implementation of "ACTIVE YOU." (L57): The authors have already presented the framework of "ACTIVE YOU" in this paper, but if we can add the relevant theories and descriptions of the planning and implementation of "ACTIVE YOU," it will be more practical for this study to provide a reference for related researchers. - We agree that some addtional references can be integrated across the introduction as justification for the ACTIVE YOU development and implementation. Please see the highlighted track changes (line 26-29; lines 39- 42).
- (L57): The authors hope to strengthen the basis of this paper by giving examples, but the existing questions alone cannot support the importance of this paper, so it is suggested that the authors can cite more conclusions from existing related studies to support the importance of this study, which can enhance the value of this study more obviously. -Yes there is existing literature focused on teacher beliefs and self-efficacy, which has now been integrated into the manuscript to strengthen the argument, beyond specific examples, which were requested by the editor on our first submission.Lines 60 - 83 have been updated to provide addtional references and support for the development of ACTIVE YOU.
-
(L114): The authors present the TIDieR Intervention Checklist in a tabular format, however, the authors could further explain the correspondence between "HOW" and "What," and the materials and procedures used in this study are the focus of the study implementation. For example, the authors used "wearables devices (accelerometers)" as the source of data collection. What is the specific information that can be collected, and what is the baseline of this study? These are the primary basis for presenting the effectiveness of this study. - Sure, happy to revise the text on the TIDieR checklist. Review 1 requested that we be more concise on the checklist, so have carefully considered the level of details for each section and prioritzed the realvance of information for replication by other scholars and teacher educator interested in implementing such a program.
-
(L168): The Data Collection Procedures section is an essential part of this study. According to the information provided by the authors, the participants were asked to wear for 14 days (7 days early in the semester and 7 days later in the semester).
The authors provide information that the participants were asked to wear for 14 days (7 days early in the semester and 7 days later in the semester). Can the authors further explain why this arrangement and the detailed implementation procedures of the study, and at the same time, the linkage between this study and the course content can also be explained through understanding the intervention of this study will help the reader to have a more detailed understanding of this study, we look forward to seeing complete content from the authors. - Revised an updated the TIDieR table. Thank you.